# Influence of patient isolation due to colonization with multidrug-resistant organisms on functional recovery after spinal cord injury

**Peter Prang, Christian Schuld, Ruediger Rupp, Cornelia Hensel, Norbert Weidner** *

Spinal Cord Injury Center, Heidelberg University Hospital, Heidelberg, Germany

* Norbert.weidner@med.uni-heidelberg.de

## Abstract

### Study design

Chart reviews were combined with neurological and functional outcome data obtained from the prospective European Multicenter Study on Spinal Cord Injury (EMSCI, www.emsci.org).

### Objectives

To determine if strict physical isolation of multidrug-resistant organisms (MDRO)-positive patients negatively affects neurological recovery and functional outcome in the first year after acute spinal cord injury (SCI).

### Setting

SCI Center Heidelberg University Hospital.

### Methods

Individuals with acute (< 6 weeks) traumatic or ischemic SCI were included. During primary comprehensive care, isolated MDRO-positive patients (n = 13) were compared with a MDRO-negative control group (n = 13) matched for functional (Spinal Cord Independence Measure–SCIM) and neurological impairment (motor scores based on the International Standards for Neurological Classification of Spinal Cord Injury—ISNCSCI) at an early stage up to 40 days after SCI. SCIM scores and motor scores were obtained at 12 weeks (intermediate stage) and 24 or 48 weeks (late stage) after SCI.

### Results

Isolated MDRO-positive (median duration of hospitalization: 175 days, 39% of inpatient stay under isolation measures) and non-isolated MDRO-negative (median duration of hospitalization: 161 days) patients showed functional and neurological improvements, which were not statistically different between groups at the intermediate and late stage.

**Data Availability Statement:** All relevant data are within the manuscript and its Supporting Information files.

**Funding:** The authors received no specific funding for this work.

**Competing interests:** The authors have declared that no competing interests exist.

## Conclusion

Prolonged isolation due to MDRO colonization for over a third of the inpatient comprehensive care period does not appear to impair neurological recovery and functional outcome within the first year after SCI.

## Introduction

Colonization with multidrug-resistant organisms (MDRO) represents a growing problem in spinal cord injury (SCI) centers and other hospital care facilities as well. Besides methicillin resistant Staphylococcus aureus (MRSA), vancomycin-resistant Enterococcus (VRE) particularly the prevalence of multidrug-resistant gram negative bacteria (MRGN) producing extended-spectrum beta-lactamase (ESBL) is still increasing [1]. In Germany, MRGN bacteria are distinguished as 3-MRGN or 4-MRGN depending on their grade of antibiotic resistance. 4-MRGN bacteria are characterized by a resistance against 4 out of 4 groups of antibiotics (ureidopenicillins, third/fourth generation cephalosporins, carbapenems, quinolones) [2]. Many individuals with SCI have received initial intensive care unit treatment, which by itself increases the risk for MDRO colonization [3]. MRSA was identified as the most common MDRO colonizing patients with SCI [4]. One study reported a prevalence rate of 39% in MRSA colonization on admission in SCI units [5]. In a North American acute rehabilitation unit the prevalence rate of MRSA in patients with traumatic brain injury, fractures and ambulation dysfunction at the time of admission increased from 5% to 12% between 1987 and 2000 [6]. Numerous risk factors for colonization with MRSA have been identified [7]. Frequently, several of these conditions namely chronic skin conditions, tracheostomy with mechanical ventilation, antibiotic therapy, and high comorbidity are present in individuals with SCI. Therefore, appropriate strategies to prevent and control colonization with MDROs are pursued [8]. Beside screening and chemical decolonization, a strict isolation of colonized patients is recommended [9]. This isolation usually interferes with rehabilitative interventions such as physical or occupational therapy, which may have a negative impact on clinical outcome. Furthermore, isolated individuals are deprived from social contacts [10].

Thus, the impact of colonization with MDRO and subsequent patient isolation on the rehabilitation outcome is of particular interest. A number of studies have focused on patient perception and satisfaction of treatment during contact isolation. Some studies indicated that treatment satisfaction of patients and caregivers was not altered by contact isolation, whereas other studies reported more frequent complaints related to the hospitalization, communication with staff and negative perception of treatment [11–15]. A study with ischemic and hemorrhagic stroke survivors demonstrated that an inferior functional and morbidity status on admission rather than patient isolation-related measures such as fewer rehabilitative interventions contributed to the less favorable outcome of MDRO-positive patients [16]. The length of stay of MDRO-positive neurological patients in rehabilitation units was prolonged compared to MDRO-negative patients [16, 17].

As of now there are no studies available, which have investigated the effects of isolation due to MDRO colonization on functional outcome after SCI. Therefore, the aim of this study was to compare the outcome of MDRO-positive patients related to the ability to perform activities of daily living with those of a matched non-isolated MDRO-negative control group during the first year after SCI. We hypothesized that the strict isolation of MDRO-positive individuals with SCI leads to an inferior functional outcome.

## Methods

### Study setting and participants

All individuals enrolled in this matched cohort study suffered from acute traumatic or ischemic SCI and were treated in the Spinal Cord Injury Center at Heidelberg University Hospital between 07/2002 and 03/2016. The site is specialized in the acute and chronic care of individuals with SCI. In Germany, patients with acute SCI are transferred to dedicated SCI centers as soon as surgical interventions (spinal decompression/stabilization) have been completed. There, acute care and rehabilitative interventions—termed comprehensive SCI care—are combined for the complete inpatient stay.

All participants were assessed within the framework of the prospective European Multicenter Study on Spinal Cord Injury (EMSCI, www.emsci.org) in respect to 1) neurological outcome according to the International Standards for Neurological Classification of Spinal Cord Injury (ISNCSCI) and 2) functional outcome by measuring the independence in activities of daily life with the Spinal Cord Independence Measure (SCIM) at defined time points (see below) during the first year after injury [18]. Ethical approval was granted by the Ethics Committee of the Medical Faculty, University of Heidelberg (S-188/2003). Written informed consent was given by each participant. Information regarding the MDRO status and duration of isolation was retrieved from medical records retrospectively. Respective data were accessed and analyzed only in a fully anonymized fashion.

Within the EMSCI project a history of dementia or severe reduction of cognition, peripheral nerve lesions above the level of lesion including polyneuropathy or severe traumatic brain injury represent exclusion criteria. For every patient enrolled in the present study inclusion criteria were as follows: a complete data set of neurological (ISNCSCI) and functional assessments (SCIM) from 3 different time points had to be available: 1) **Early stage**–either from day 1–15 after injury (*very acute* according to the EMSCI protocol) or day 15–40 (*acute I* according to the EMSCI protocol), 2) **intermediate stage**– 12 weeks after injury (*acute II* according to the EMSCI protocol) or 3) **late stage**–either 24 (*acute III* according to the EMSCI protocol) or 48 weeks after injury (*chronic* according to the EMSCI protocol). Additional inclusion criteria for the isolated MDRO-positive cohort were the presence of prolonged strict physical isolation measures for at least 25 days due to colonization with MRSA or 4-MRGN during inpatient rehabilitation. The exact cut-off for the minimum duration of isolation—25 days–was set arbitrarily based on the assumption that only a prolonged duration of isolation measures would impact the rehabilitative outcome. In the MDRO-negative non-isolated cohort isolation measures were not identifiable according to the review of respective medical records. Isolation measures for inpatients colonized with MRSA or 4-MRGN remained unchanged over the study period.

### Multidrug-resistant organisms and patient isolation measures

On admission all patients were systematically screened for MRSA-, VRE- and MRGN-colonization. As soon as MRSA and 4-MRGN colonization was confirmed, patients were placed in single-, two- or three-bed rooms together with up to 2 patients colonized with identical germs. Therapists were prompted to perform hand disinfection, to wear mouth/nose protection and gloves and gowns before starting rehabilitative interventions. Appropriate cleaning and disinfection of training devices such as training benches were conducted. Precautions to prevent spread of MRSA and 4-MRGN were taken according to the guideline for isolation precautions of Centers for Disease Control and Prevention, U.S. Department of Health & Human Services (https://www.cdc.gov/mrsa/healthcare/clinicians/precautions.html) and recommendations from the Robert Koch Institute, Germany [19].

Rehabilitative interventions can be categorized in analogy to the International Classification of Functioning, Disability and Health (ICF) (http://www.who.int/classifications/icf/en/) into the levels body structures, basic activities and complex activities [20]. Isolation-related measures mainly restricted the use of a variety of machines required for interventions at the body structure/function (e.g. machine-based endurance and muscular strength training) and activity level (e.g. body weight supported treadmill or practicing the transfer from the wheelchair into a car). Substitutions at the body structure/function (e.g. resistance bands or weights, arm/leg cycling training brought into the patient room) and basic activity level (e.g. parallel bars for walking training or arm suspension devices for hand/arm use in a room reserved for MDRO patients) were available for isolated patients instead. At the complex activity level, patient training, e.g. related to household tasks in the patient adapted kitchen, was omitted. Freed-up therapy slots were filled with rehabilitative interventions, which could be applied despite isolation.

## Standardized assessments

The standardized neurological examination was conducted according to International Standards for Neurological Classification of Spinal Cord Injury (ISNCSCI) [21] by trained assessors [22] supported by computer-assisted scaling, scoring and classification including the American Spinal Injury Association Impairment Scale (AIS). All ISNCSCI data were re-classified according to the updated 7th edition [21] as described [23]. Impairment of motor function was examined by upper extremity motor score (UEMS) and lower extremity motor score (LEMS) performed by manual testing of five key muscles of each limb (0 = total paralysis to 5 = active movement to full resistance). The UEMS/LEMS represents the bilateral aggregate motor score of 5 upper/lower limb myotomes with a maximum score of 50.

Functional recovery of individuals was examined by measuring the independence in activities of daily life with the Spinal Cord Independence Measure (SCIM) [24]. Individuals included before December 2007 were assessed using the SCIM II [25], those enrolled thereafter with the SCIM III [26]. In the SCIM III single items were changed or deleted, but the subscale and total scores remained unchanged [27]. Three sub-categories are included in the SCIM: self-care (subscore, 0–20), respiration and sphincter management (subscore, 0–40), and mobility (subscore, 0–40). The total SCIM score ranges from 0 to 100.

Furthermore, the Walking Index for Spinal Cord Injury II (WISCI II) was performed. The WISCI II quantifies the dependence on walking aids and/or physical assistance using an ordinal scale ranging from 0 to 20. The maximal score is assigned if the participant is able to walk 10m without braces, supports or human aids [28, 29]. All functional and neurological tests were collected within the prospective EMSCI study.

## Statistical analysis

All statistical analyses were performed using Statistica® software version 9 (StatSoft Inc., OK, USA). People with SCI were divided in two groups (MDRO-positive and MDRO-negative patients). Each included MDRO-positive SCI patient was matched with a non-isolated MDRO-negative patient who was identified in the EMSCI database of the Heidelberg SCI Center (S1 Table).

Cohorts consisted of either non-isolated MDRO-negative or isolated MDRO-positive inpatients. Once eligible MDRO-positive inpatients were identified, they were matched according to their neurological (ISNCSCI) and functional (SCIM) status at any early stage after admission (from 1–40 days after injury) with non-isolated MDRO-negative patients. Matching criteria were: Similar UEMS and LEMS scores ± 9 points and SCIM score ± 9 points in the

initial assessment. These data are presented as mean ± standard error of the mean. Characteristics of included participants are presented as median and interquartile range.

Examinations comprised the SCIM, UEMS and LEMS. Participants were compared for differences between the two groups using a Wilcoxon signed rank test for matched samples. Probability values less than 0.05 were considered statistically significant.

## Results

### Participants

From a total of 906 acute SCI patients admitted between 07/2002 and 03/2016 to the SCI Center at Heidelberg University Hospital 381 individuals were included in the EMSCI project. Of these, 323 individuals with three complete SCIM assessments were identified.

Within this sample, 43 MDRO positive subjects were identified. Another 30 patients had to be excluded: 4 patients with VRE were excluded because of less strict and varying isolation measures over the years and 16 patients with 3-MRGN, who did not undergo strict isolation measures. Furthermore, 2 patients with 4-MRGN and 8 patients with MRSA were excluded because their duration of isolation was less than 25 days. A total of 13 patients were isolated due to a positive MDRO status (MRSA or 4-MRGN) for at least 25 days (Table 1, S1 Table). All MDRO-positive patients spent 39% of their total stay in the hospital under isolation measures. The length of stay did not yield a significant difference between the MDRO-positive and the MDRO-negative group. The majority of the included MDRO-positive patients were MRSA-positive (n = 8), the five remaining were 4-MRGN-positive.

### Neurological recovery and functional improvements over time

Individuals in both groups started with comparable neurological (UEMS, LEMS) and functional outcome parameters (SCIM) (Table 1, Fig 1 and S1 Table).

**Table 1. Characteristics of included participants.**

| Variables | | MDRO-positive | MDRO-negative | p-value |
|---|---|---|---|---|
| Tetraplegia [N] | | 9 (69%) | 10 (77%) | 0.65 |
| Paraplegia [N] | | 4 (31%) | 3 (23%) | |
| AIS [N] | | | | 0.68 |
| | A | 7 (54%) | 6 (46%) | |
| | B | 2 (15%) | 1 (8%) | |
| | C | 4 (31%) | 5 (38%) | |
| | D | 0 (0%) | 1 (8%) | |
| Motor complete [N] | | 9 (69%) | 7 (54%) | 0.42 |
| Motor incomplete [N] | | 4 (31%) | 6 (46%) | |
| Gender [N] | Female | 4 (31%) | 5 (38%) | 0.68 |
| | Male | 9 (69%) | 8 (62%) | |
| Upper extremity motor score [points, 0–50] | | 19 (12–50) | 18 (12–50) | 0.60 |
| Lower extremity motor score [points, 0–50] | | 0 (0–11) | 0 (0–10) | 0.69 |
| Spinal Cord Independence Measure [points, 0–100] | | 5 (0–13) | 10 (0–13) | 0.30 |
| Median Age [years] | | 59.0 (39.0–64.0) | 45.0 (36.0–58.0) | 0.75 |
| Median Length of stay [days] | | 175.0 (161.0–208.0) | 161.0 (133.0–244.0) | 0.09 |
| Median interval from onset SCI until admission at SCI center [days] | | 18.0 (12.0–23.0) | 13.0 (6.0–24.0) | 0.38 |
| Median Length of isolation [days] | | 72.0 (53.0–137.0) | | |
| Median interval from onset SCI until start of isolation measures [days] | | 46.0 (32.0–60.0) | | |

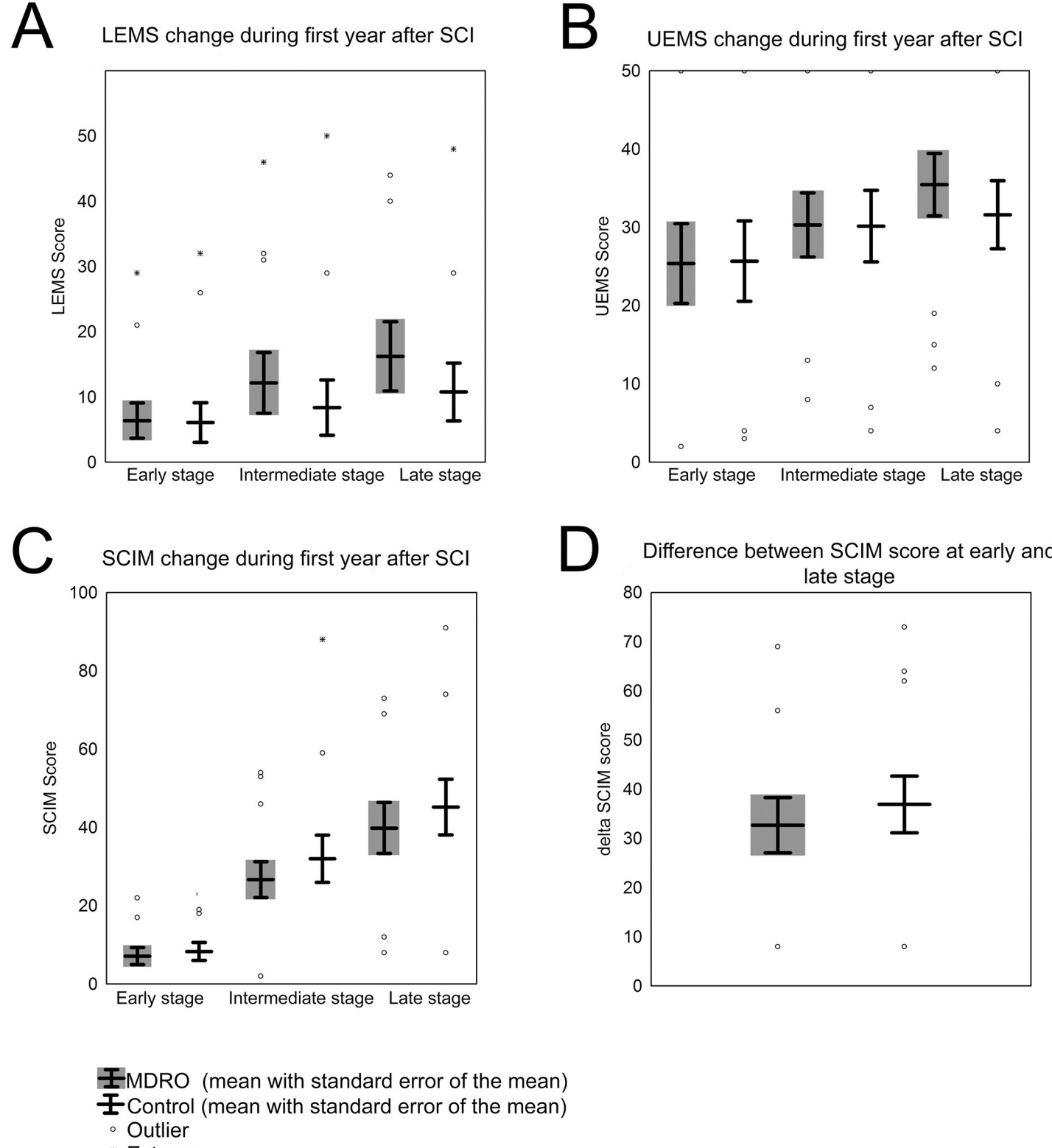

**Fig 1. Motor recovery and functional improvement after SCI.** Boxplots show (A) Lower Extremity Motor Score (LEMS), (B) Upper Extremity Motor Score (UEMS) and (C) Spinal Cord Independence Measure (SCIM) of the MDRO-positive patients compared with matched MDRO-negative patients at the early, intermediate and late stage. (D) Difference of the total SCIM score between late and early stage. Dots represent outlier ($>$1.5 times standard error), asterisks extreme outlier ($>$ 2 x 1.5 times standard error).

**Table 2. Assessment interval related to SCI onset.**

| Variables | MDRO-positive | MDRO-negative | p-value |
|---|---|---|---|
| Median interval SCI onset to UEMS/LEMS assessment—early stage [days] | 29.0 (20.0–39.0) | 29.0 (22.0–33.0) | 0.53 |
| Median interval SCI onset to UEMS/LEMS assessment—intermediate stage [days] | 81.0 (79.0–87.0) | 84.0 (83.0–84.0) | 0.76 |
| Median interval SCI onset to UEMS/LEMS assessment—late stage [days] | 308.0 (173.0–350.0) | 302.0 (159.0–359.0) | 0.46 |
| Median interval SCI onset to SCIM assessment—early stage [days] | 24.0 (19.0–33.0) | 28.0 (28.0–32.0) | 0.86 |
| Median interval SCI onset to SCIM assessment—intermediate stage [days] | 84.0 (84.0–86.0) | 84.0 (84.0–84.0) | 0.83 |
| Median interval SCI onset to SCIM assessment—late stage [days] | 300.0 (169.0–341.0) | 300.0 (160.0–310.0) | 0.53 |

Over time, neurological recovery assessed with the UEMS and LEMS was not significantly different between groups at any stage investigated (Fig 1A and 1B). Moreover, the evolution of the functional outcome measured with the SCIM total score did not differ between MDRO-positive and matched control SCI patients (Fig 1C and 1D). Both, the absolute SCIM score at the late stage (Fig 1C) and the change of the SCIM score between the early and the late stage (Fig 1D) were not significantly different between groups (p = 0.65 and p = 0.75 respectively). Of note, neurological and functional assessment time points—referenced to the date of injury —did not differ in both cohorts (Table 2).

Analysis of the three SCIM sub-categories–self care (Fig 2A), respiration/sphincter management (Fig 2B) and mobility (Fig 2C)—did not show any significant difference between the groups at any stage (Fig 2). The self care subscore showed a trend towards higher functional outcome in matched MDRO-negative patients at the intermediate and late stage but without significant difference (p = 0.07 and p = 0.21, respectively).

Only 4 MDRO-positive patients were motor incomplete, which precluded a meaningful statistical analysis of this subgroup. We compared mobility and walking related functional outcomes (SCIM, WISCI II) in matched incomplete SCI individuals at the late stage, which does not suggest a better outcome in favor of the MDRO-negative individuals (S2 Table).

## Discussion

Based on our findings we cannot confirm that isolation measures implemented after colonization with MDRO impair the neurological and more importantly functional status up to 1 year after injury. We found no significant differences in the SCIM scores or SCIM subscores between the isolated MDRO-positive cohort and the non-isolated control group assessed in an intermediate (12 weeks) or late stage (24 or 48 weeks). In accordance with these results we also did not find any effect on SCIM score difference (SCIM score in the late phase minus SCIM in the early phase) representing functional improvements over 24 or 48 weeks after injury in both groups.

To our knowledge, this is the first study which assesses the effect of isolation measures due to MDRO colonization on functional recovery after acute SCI. Outcome data from the first year after SCI are of high quality since neurological (ISCNSCI) and functional (SCIM) assessments were obtained and documented by trained assessors at predefined time points after injury. Moreover, matching of MDRO-positive and negative patients based on their neurological status (upper and lower extremity motor score) generated comparable starting conditions in respect to the extent of neurological dysfunction, which allowed to analyze the impact of isolation measures more rigorously.

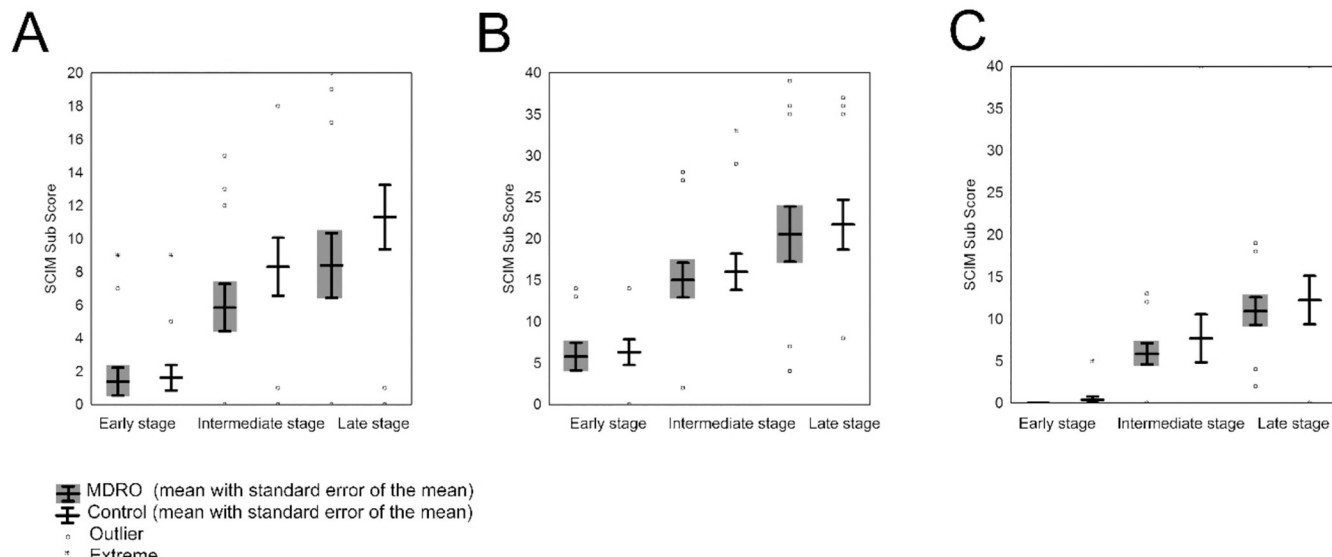

**Fig 2. Functional outcome determined by SCIM assessment within the first year after injury.** Changes of sub-SCIM items (A) self-care, (B) respiration & sphincter management and (C) mobility at the early, intermediate and late stage. Dots represent outlier (>1.5 times standard error); Asterisks represent extreme outlier (>2 x 1.5 times standard error).

Our findings are in line with a previous study, where functional outcome was investigated in MDRO-positive patients suffering from critical illness polyneuropathy, ischemic and hemorrhagic stroke [30]. In this patient cohort, morbidity and functional level were low on admission. However, the change of the functional status measured with the Barthel index on admission and before discharge was not different between MDRO-positive and MDRO-negative patients.

In addition to the similar functional outcome we did not observe a significant difference in the overall length of stay. A monocentric study in a SCI center in the United Kingdom identified an impressive difference in respect to the length of stay, where MDRO-positive patients stayed in the hospital for more than one year (412 days), whereas their matched MDRO-negative cohort was treated for around half a year (187 days) [17, 31]. However, their patient cohorts are not really comparable to our study group. The range of intervals from the date of injury until hospital admission varied vastly from 2 to 304 days, which clearly indicates that both acute and chronic SCI patients were included in their study. In the present study, only acute SCI patients admitted for primary comprehensive SCI care were included, which is reflected in the much shorter interval from date of injury until admission (range 6 to 24 days).

The SCIM subitem self care showed a trend towards better outcome in the matched MDRO-negative group. Self care includes activities of daily living such as feeding, bathing, dressing and grooming. These skills are at least partly practiced using specific training devices, which were not accessible for MDRO-positive patients due to isolation measures. In contrast, respiration and sphincter management can easily be practiced in the patient's room. As a consequence, the corresponding outcome score (SCIM sub-category respiration/sphincter) was not found to be different in isolated versus non-isolated SCI patients. A potential confounding factor, which could at least partly explain the trend towards better functional outcome in the MDRO negative group was the lower average age in this group. It is known that with older age functional outcome as assessed by the SCIM scores is inferior compared to younger SCI patients with similar neurological dysfunction early after injury [32, 33].

The assumption that incomplete SCI patients, which may require more exposure to activity-focused interventions outside of the patient's room (e.g. robotic assisted locomotion training), are more prone to detrimental isolation effects, could not be verified in the present study.

Isolation measures can affect the psychological well-being of patients. Depression and anxiety have been described to be more pronounced in isolated patients compared to non-isolated patients [34, 35]. The psychological impact of isolation may depend on specific risk factors like patients' age and may be more pronounced in older individuals. In our study isolation measures may have affected psychological well-being. However, validated scales to evaluate mood, depression or anxiety were not applied.

The study is limited by the small sample size which is not large enough to ensure adequate statistical power. The small sample size becomes even more prominent in the subgroup of incomplete SCI patients, where negative effects of strict isolation measures are potentially more likely. Thus, a moderate change in respect to inferior functional outcome due to patient isolation may have been missed. The total observation period of this study was rather prolonged [14 years], which might challenge the validity of the results. However, only patients colonized with MRSA and 4-MRGN were included. In these cases, isolation measures did not change over the observation period of this study. Alternatively, neurological and functional outcome may have shifted over the years due to changes in standards of SCI care. However, according to a recent respective analysis of the EMSCI database covering the period from 2002 until 2019 this is not the case (personal communication, Armin Curt, Zurich, Switzerland). Medical records did not contain information regarding the room occupancy of each patient. This variable could have affected inpatients' experience, coping strategies and ultimately rehabilitation outcomes. Rehabilitative interventions were not documented in terms of quantity and quality. A standardized system to record rehabilitative interventions has been developed recently within EMSCI, which was not yet available for patients included in the present study [36]. Therefore, the exact modification of physical and occupational therapy induced by the isolation measures cannot be verified.

Overall, this monocentric study suggests that strict isolation measures do not affect functional outcome in people with SCI within the first year after injury. Our results should give more confidence to rehabilitation experts, who have to negotiate competing interests of optimal rehabilitation outcome versus adequate hygiene standards in the hospital setting. Of course, these findings need to be replicated in a prospective study, which also records the quality and quantity of interventions in isolated versus non-isolated patients.

## Supporting information

**S1 Table. Neurological and functional status of matched participants at early stage.** (DOCX)

**S2 Table. SCIM and WISCI II of matched incomplete patients at late stage.** (DOCX)

## Acknowledgments

We thank all individuals and all centers of the EMSCI network (https://emsci.org). Furthermore, we thank Elisabeth Nowak, Anne von Reumont and Brigitte Nussbaum for excellent support.

## Author Contributions

**Conceptualization:** Peter Prang, Christian Schuld, Norbert Weidner.

**Data curation:** Christian Schuld.

**Formal analysis:** Peter Prang, Christian Schuld, Ruediger Rupp.

**Investigation:** Peter Prang, Christian Schuld, Ruediger Rupp, Cornelia Hensel.

**Methodology:** Peter Prang, Christian Schuld, Ruediger Rupp, Norbert Weidner.

**Supervision:** Norbert Weidner.

**Writing – original draft:** Peter Prang, Christian Schuld, Cornelia Hensel, Norbert Weidner.

**Writing – review & editing:** Ruediger Rupp, Norbert Weidner.

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
