## [Decision Letter · Decision Letter 0]

26 Jan 2021

PONE-D-20-22478

INFLUENCE OF PATIENT ISOLATION DUE TO COLONIZATION WITH MULTIDRUG-RESISTANT ORGANISMS ON FUNCTIONAL RECOVERY AFTER SPINAL CORD INJURY

PLOS ONE

Dear Dr. Weidner,

Thank you for submitting your manuscript to PLOS ONE. After careful consideration, we feel that it has merit but does not fully meet PLOS ONE’s publication criteria as it currently stands. Therefore, we invite you to submit a revised version of the manuscript that addresses the points raised during the review process.

Two experts in the field have reviewed your manuscript and have several comments and suggestions for revision. Please carefully address these. Also, please note that one reviewer suggests that this may be condensed into a brief report. PLOS ONE does not usually publish brief reports, so you may disregard this comment. 

We look forward to receiving your revised manuscript.

Kind regards,

Susan Hepp

Academic Editor

PLOS ONE

Journal Requirements:

Reviewers' comments:

Reviewer's Responses to Questions

**Comments to the Author**

1. Is the manuscript technically sound, and do the data support the conclusions?

Reviewer #1: Partly

Reviewer #2: Yes

2. Has the statistical analysis been performed appropriately and rigorously? 

Reviewer #1: No

Reviewer #2: Yes

3. Have the authors made all data underlying the findings in their manuscript fully available?

Reviewer #1: Yes

Reviewer #2: Yes

4. Is the manuscript presented in an intelligible fashion and written in standard English?

Reviewer #1: Yes

Reviewer #2: Yes

5. Review Comments to the Author

Reviewer #1: Background:

1. There have been several studies in general acute care patient populations evaluating the impact of contact isolation on outcomes such as patient satisfaction and other patient centered outcomes that could be further described.

Methods:

1. Include the actual study design in the methods. It appears to be a matched cohort study design where the cohort is matched on exposure (isolation or non-isolation) and the outcome assessed is functional outcomes.

2. Study Design and setting: Inclusion criteria should vary for those in isolation vs those not in isolation. As currently written, inclusion criteria suggest that those with MDROs and placed in contact precautions were included. This is only relevant for those who had MDRO and isolated, not the patients who didn’t have an MDRO and were not isolated. Clarify inclusion criteria for each group.

3. The enrollment period includes 14 years which is very long given the relatively low sample size. Changes in the isolation protocols and procedures for patients that may have happened during this time period is not adequately discussed.

4. There is not enough explanation as to why only those with more than 25 days of isolation are included. Given the small sample size, the number of patients that were excluded because of this criterion should be presented along with a stronger rationale especially if this resulted in a significant number of exclusions.

5. The main hypothesis is that individuals isolated because of MDRO infections recovery will suffer in part due to their isolation. However, isolation rooms potentially held 2-3 people per room. Information on how many patients had roommates in isolation and the potential affect this could have had on the outcomes should be discussed.

Results:

1. Only 8% of the spinal cord patients in the program are included in this analysis, it seems unlikely that the MDRO rate was this low. More explanation of how many patients were excluded and the reasons why need to be discussed.

2. The tables could be simplified. Showing the matched pairs information in table 1 can be eliminated and just show the percentages and means/medians for each variable and add it to Table 2. Table 4 can be eliminated.

3. Table 2: Although these are small sample sizes, including percentages would be helpful for ease of viewing and interpretation.

4. Table 2: Include p-values for all variables, not just the continuous variables.

This paper could be shortened to a brief report.

Reviewer #2: Authors proposed a very relevant issue in the field of neurorehabilitation, the relevance of patients isolation due to colonization with multidrug-resistant bacteria. The review charts of 906 acute SCI patients identifying a total of 13 individuals isolated due to a positive MDRO for at least 25 days. With the aim to determine if strict physical isolation of multidrug-resistant organisms (MDRO)- positive patients negatively affects neurological recovery and functional outcome in the first year after acute spinal cord injury (SCI) Authors compared MDRO positive patients versus MDRO negative patients matched for functional (Spinal Cord Independence Measure – SCIM) and neurological impairment (motor scores based on the International Standards for Neurological Classification of Spinal Cord Injury - ISNCSCI). SCIM scores and motor scores were obtained at 12, 24 or 48 weeks after SCI. Authors found that functional and neurological improvements, observed for both groups were not statistically different between groups at the intermediate and late stage. Authors conclude that prolonged isolation due to MDRO colonization does not appear to impair neurological recovery and functional outcome within the first year after SCI.

I think that the study has several limitations that the Authors correctly report in the discussion: small sample size; mood, depression and anxiety not evaluated; no description of the rehabilitation interventions performed. The study is certainly interesting, however the small sample certainly alters the statistical data as evidenced also by the fact that there is a tendency in the negative MDRO group to reach better values than in the positive MDRO group. Furthermore, there are factors such as age that authors should discuss. There is also difficulty in the methodological part in understanding how the two groups of patients were selected and matched. Suggestions: review the methods section (report here the methods/study design and data relating to the review of the 906 medical records), I would better describe the selection method of the control group; broaden the discussion by discussing the different factors that may have affected the final results.

6. PLOS authors have the option to publish the peer review history of their article (what does this mean?). If published, this will include your full peer review and any attached files.

Reviewer #1: No

Reviewer #2: **Yes: **Bartolo Michelangelo

---

## [Author Response · Author response to Decision Letter 0]

5 Feb 2021

Reviewer #1:

Background: 1. There have been several studies in general acute care patient populations evaluating the impact of contact isolation on outcomes such as patient satisfaction and other patient centered outcomes that could be further described.

Response: Respective information has been added to the Introduction.

Methods:

 1. Include the actual study design in the methods. It appears to be a matched cohort study design where the cohort is matched on exposure (isolation or non-isolation) and the outcome assessed is functional outcomes.

Response: As suggested this information has been added to the Methods section.

2. Study Design and setting: Inclusion criteria should vary for those in isolation vs those not in isolation. As currently written, inclusion criteria suggest that those with MDROs and placed in contact precautions were included. This is only relevant for those who had MDRO and isolated, not the patients who didn’t have an MDRO and were not isolated. Clarify inclusion criteria for each group.

Response: We clarified the inclusion criteria: Additional inclusion criteria for the isolated MDRO-positive cohort were the presence of prolonged strict physical isolation measures (minimum of 25 days) due to MDRO colonization during inpatient rehabilitation as determined by review of medical records. In the MDRO-negative non-isolated cohort isolation measures were not identifiable according to the review of respective medical records.

3. The enrollment period includes 14 years which is very long given the relatively low sample size. Changes in the isolation protocols and procedures for patients that may have happened during this time period is not adequately discussed.

Response: Absolutely valid point. We have added respective information to the Methods section and addressed it as a further limitation. We only included patients with MRSA and 4-MRGN, which were under identical isolation measures over the complete study observation period as described in the methods section. 

4. There is not enough explanation as to why only those with more than 25 days of isolation are included. Given the small sample size, the number of patients that were excluded because of this criterion should be presented along with a stronger rationale especially if this resulted in a significant number of exclusions.

Response: The exact cut-off for the minimum duration of isolation - 25 days – was set arbitrarily based on the assumption that only a prolonged duration of isolation measures would impact the rehabilitative outcome. The patient selection process was as follows. Patient records between 2002 and 2016 were screened for colonization with MRSA, 4-MRGN, 3-MRGN and VRE. From these initial pre-selection, 43 MDRO positive subjects were identified, which had complete neurological and functional datasets as pointed out in the manuscript. Of those 43 patients, 30 patients had to be excluded: 4 patients with VRE were excluded because of less strict and varying isolation measures over the years and 16 patients with 3-MRGN, which did not undergo strict isolation measures. Furthermore, 2 patients with 4-MRGN and 8 patients with MRSA were excluded because their duration of isolation was less than 25 days. Finally, 13 patients with MRSA or 4-MRGN colonization, consecutive strict isolation measures and a complete neurological and functional dataset went into the comprehensive analysis. The respective information has been added to the Methods section.

5. The main hypothesis is that individuals isolated because of MDRO infections recovery will suffer in part due to their isolation. However, isolation rooms potentially held 2-3 people per room. Information on how many patients had roommates in isolation and the potential affect this could have had on the outcomes should be discussed.

Response: Another excellent point. We added this aspect in the Discussion: Medical records did not contain information regarding the room occupancy of each patient. This variable could have affected inpatients’s experience, coping strategies and ultimately rehabilitation outcomes.

Results:

1. Only 8% of the spinal cord patients in the program are included in this analysis, it seems unlikely that the MDRO rate was this low. More explanation of how many patients were excluded and the reasons why need to be discussed.

Response: See answer above.

2. The tables could be simplified. Showing the matched pairs information in table 1 can be eliminated and just show the percentages and means/medians for each variable and add it to Table 2. Table 4 can be eliminated.

Response: We changed the manuscript accordingly. The information of table was condensed and added to table 2 as suggested. Table 1 and 4 are now provided as supplementary information. 

3. Table 2: Although these are small sample sizes, including percentages would be helpful for ease of viewing and interpretation.

Response: Done.

4. Table 2: Include p-values for all variables, not just the continuous variables.

Response: Done

This paper could be shortened to a brief report.

Response: We shortened the manuscript by moving 2 tables to the supplementary data. Beyond this, we think the length of the manuscript and description of the findings as it stands right now is required in order to support the reader in understanding results and conclusions.

Reviewer #2: Authors proposed a very relevant issue in the field of neurorehabilitation, the relevance of patients isolation due to colonization with multidrug-resistant bacteria. The review charts of 906 acute SCI patients identifying a total of 13 individuals isolated due to a positive MDRO for at least 25 days. With the aim to determine if strict physical isolation of multidrug-resistant organisms (MDRO)- positive patients negatively affects neurological recovery and functional outcome in the first year after acute spinal cord injury (SCI) Authors compared MDRO positive patients versus MDRO negative patients matched for functional (Spinal Cord Independence Measure – SCIM) and neurological impairment (motor scores based on the International Standards for Neurological Classification of Spinal Cord Injury - ISNCSCI). SCIM scores and motor scores were obtained at 12, 24 or 48 weeks after SCI. Authors found that functional and neurological improvements, observed for both groups were not statistically different between groups at the intermediate and late stage. Authors conclude that prolonged isolation due to MDRO colonization does not appear to impair neurological recovery and functional outcome within the first year after SCI.

I think that the study has several limitations that the Authors correctly report in the discussion: small sample size; mood, depression and anxiety not evaluated; no description of the rehabilitation interventions performed. The study is certainly interesting, however the small sample certainly alters the statistical data as evidenced also by the fact that there is a tendency in the negative MDRO group to reach better values than in the positive MDRO group. Furthermore, there are factors such as age that authors should discuss. 

Response: A confounding factor, which could at least partly explain the trend towards better functional outcome in the MDRO negative group was the lower average age in this group. It is known that with older age functional outcome as assessed by the SCIM scores is inferior compared to younger SCI patients with similar neurological dysfunction early after injury. This aspect has been added in the Discussion.

There is also difficulty in the methodological part in understanding how the two groups of patients were selected and matched. Suggestions: review the methods section (report here the methods/study design and data relating to the review of the 906 medical records), I would better describe the selection method of the control group; 

Response: Please see comments. Respective aspects have been added to the manuscript.

Broaden the discussion by discussing the different factors that may have affected the final results.

Response: The confounding factor age is discussed now. As further limitations of the study we discuss now the aspect of the prolonged study period and different rooming situations (single, 2- or 3-bed-room), which could have differently affected the impact of isolation measures.

---

## [Decision Letter · Decision Letter 1]

16 Mar 2021

INFLUENCE OF PATIENT ISOLATION DUE TO COLONIZATION WITH MULTIDRUG-RESISTANT ORGANISMS ON FUNCTIONAL RECOVERY AFTER SPINAL CORD INJURY

PONE-D-20-22478R1

Dear Dr. Weidner,

We’re pleased to inform you that your manuscript has been judged scientifically suitable for publication and will be formally accepted for publication once it meets all outstanding technical requirements.

Kind regards,

Antal Nógrádi, M.D., Ph.D., D.Sc.

Academic Editor

PLOS ONE

Additional Editor Comments (optional):

Reviewers' comments:

Reviewer's Responses to Questions

**Comments to the Author**

1. If the authors have adequately addressed your comments raised in a previous round of review and you feel that this manuscript is now acceptable for publication, you may indicate that here to bypass the “Comments to the Author” section, enter your conflict of interest statement in the “Confidential to Editor” section, and submit your "Accept" recommendation.

Reviewer #1: All comments have been addressed

Reviewer #2: All comments have been addressed

2. Is the manuscript technically sound, and do the data support the conclusions?

Reviewer #1: Yes

Reviewer #2: Yes

3. Has the statistical analysis been performed appropriately and rigorously? 

Reviewer #1: Yes

Reviewer #2: Yes

4. Have the authors made all data underlying the findings in their manuscript fully available?

Reviewer #1: Yes

Reviewer #2: No

5. Is the manuscript presented in an intelligible fashion and written in standard English?

Reviewer #1: Yes

Reviewer #2: Yes

6. Review Comments to the Author

Reviewer #1: Overall Comments: This manuscript focuses on a matched case control study comparing neurological recovery and functional outcome in MDRO-positive patients and MDRO-negative patients.

1. I would suggest using one set of terminology for the control group, either MDRO-negative patients or matched control SCI patients. Neither is particularly wrong but having one standard word choice would make the reading easier.

Results:

1. Is there information you can include on the number of rehabilitation appointments that each group received, or the average that was received? It is presumed that because the MDRO-positive group is in isolation they may have less opportunities for rehabilitation or that the quality may be poor. However, the number of rehabilitation sessions is not identified in the groups, which if the same between could help identify why there were no differences seen in neurological recovery and functional outcome. If this information is not identifiable, it should be accounted for within the limitations.

2. For table 1, please add a footnote that the motor scores and SCIM were those at onset of admission rather than just noting it in the text.

Reviewer #2: I think that after revision, now the text has improved and it is more suitable for publication. For me the paper is acceptable in this form.

7. PLOS authors have the option to publish the peer review history of their article (what does this mean?). If published, this will include your full peer review and any attached files.

Reviewer #1: No

Reviewer #2: **Yes: **Bartolo Michelangelo, MD, PhD

---

## [Editor Report · Acceptance letter]

18 Mar 2021

PONE-D-20-22478R1 

Influence of patient isolation due to colonization with multidrug-resistant organisms on functional recovery after spinal cord injury 

Dear Dr. Weidner:

I'm pleased to inform you that your manuscript has been deemed suitable for publication in PLOS ONE. Congratulations! Your manuscript is now with our production department. 

Kind regards, 

on behalf of

Prof. Antal Nógrádi 

Academic Editor

PLOS ONE